# The Effect of the Gallbladder Environment during Chronic Infection on *Salmonella* Persister Cell Formation

**DOI:** 10.3390/microorganisms10112276

**Published:** 2022-11-16

**Authors:** Juan F. González, Regan Hitt, Baileigh Laipply, John S. Gunn

**Affiliations:** 1Center for Microbial Pathogenesis, Abigail Wexner Research Institute at Nationwide Children’s Hospital, Columbus, OH 43215, USA; 2Infectious Diseases Institute, The Ohio State University, Columbus, OH 43210, USA; 3Department of Pediatrics, College of Medicine, The Ohio State University, Columbus, OH 43210, USA

**Keywords:** *Salmonella*, persisters, biofilm, bile, typhoid

## Abstract

Typhoid fever is caused by *Salmonella* enterica serovar Typhi (*S.* Typhi). Around 3–5% of individuals infected become chronic carriers, with the gallbladder (GB) as the predominant site of persistence. Gallstones (GS) aid in the development and maintenance of GB carriage, serving as a substrate to which *Salmonellae* attach and form a biofilm. This biofilm matrix protects bacteria from the host immune system and environmental stress. This shielded environment is an ideal place for the development of persister cells, a transient phenotype of a subset of cells within a population that allows survival after antibiotic treatment. Persisters can also arise in response to harsh environments such as the GB. Here we investigate if GB conditions affect the number of persisters in a *Salmonella* population. To simulate the chronic GB environment, we cultured biofilms in cholesterol-coated 96-well plates in the presence of ox or human bile. We then treated planktonic or biofilm *Salmonella* cultures with high concentrations of different antibiotics. This study suggests that biofilms provide a niche for persister cells, but GB conditions either play no role or have a negative influence on persister formation, especially after kanamycin treatment. The antibiotic target was important, as antimicrobials directed against DNA replication or the cell wall had no effect on persister cell formation. Interestingly, repeated treatment with ciprofloxacin increased the percentage of *S*. Typhimurium persisters in a biofilm, but this increase was abolished by GB conditions. On the other hand, repeated ciprofloxacin treatment of *S*. Typhi biofilms in GB conditions slightly increased the fraction of persisters. Thus, while the harsh conditions in the GB would be thought to give rise to increased persisters, therefore contributing to the development of chronic carriage, these data suggest persister cell formation is dampened in this environment.

## 1. Introduction

Typhoid fever is primarily caused by *Salmonella enterica* serovar Typhi (*S.* Typhi). Around 3–5% of individuals infected with *S.* Typhi become chronic carriers, with the gallbladder (GB) as the primary site of persistence. About 15% of patients with typhoid fever, who appear to be treated successfully, suffer from relapse [1]. In the case of chronic typhoid carriers, antibiotic treatment is often ineffective. Cholecystectomy significantly improves cure rates but is not always a viable option in endemic areas and does not guarantee clearance [2]. Gallstones (GS) aid in the development and maintenance of GB carriage, serving as a substrate to which *Salmonellae* attach and form a biofilm [3,4].

Biofilm-associated chronic infections account for more than 80% of microbial infections in the United States [5]. Aggregation in a biofilm matrix provides bacteria with a physical barrier that protects them from the host immune system and environmental stress, including antibiotics [6]. This protection, along with antibiotic resistance that arises from heritable mutations, is a reason for why these infections are notoriously hard to treat.

Persister cells are a less studied cause for the recalcitrance of biofilm-associated chronic infections. The existence of persister cells was discovered soon after antibiotics became widely used [7]. Persistence is a transient phenotype of a subset of cells within a microbial population that allows for increased survival after antibiotic treatment [8]. These cells are genetically identical to the wild-type population but, even though susceptible to antibiotics, a small fraction of these tolerant bacteria will survive antimicrobial treatment by entering a dormant state [9,10]. The regrowth of persisters gives rise to a new antibiotic-sensitive population with a similar subpopulation of tolerant cells [11]. The capacity of biofilms to shelter persister cells is suggested as a primary reason for why eliminating a chronic biofilm infection is so difficult [12]. The mechanisms for persister formation are a matter of heated debate and include toxin-antitoxin modules [13,14], intracellular ATP levels [15,16], and slow growth [17,18]. One thing is certain, persisters form stochastically but most notably arise under various stresses including not only antimicrobials, but also heat, acid, oxidative, and hyperosmotic stress [9]. The GB is a harsh environment, as bile can disrupt cellular membranes, damage DNA, and denature proteins [19,20]. Even though *Salmonella* has developed mechanisms to survive under these conditions [20], this environment seems to be the ideal location for persisters to arise. Because *S*. Typhi is a human-restricted pathogen and there are no known environmental reservoirs, in certain populations carriers are responsible for much of spread of this disease [4]. Understanding the mechanisms that allow the establishment and persistence of chronic typhoid in the GB is a critical step in the development of new therapeutic and treatment strategies.

## 2. Materials and Methods

### 2.1. Bacterial Strains and Growth Conditions

*Salmonella* strains are listed in Appendix A. Strains were streaked on Luria–Bertani (LB; 21284, Fisher Scientific, Waltham, MA, USA) agar plates and incubated at 37 °C overnight. Single colonies were used to start overnight (O/N) liquid cultures. Planktonic cells were grown at 37 °C on a rotating drum in tryptic soy broth (TSB; 211822, BD, Franklin Lakes, NJ, USA). To simulate GB conditions, media were supplemented with two different types of bile: commercially available ox bile (OB, 195498, MP Biomedical, Santa Ana, CA, USA) and human bile (HB) from our cholecystitis patient collection. Human bile was collected under an approved Institutional Review Board protocol (2018H0104/IRB18-01226) from consenting patients undergoing elective cholicysectomy.

### 2.2. Minimum Inhibitory Concentration

Minimum inhibitory concentrations for our *S*. Typhimurium and *S*. Typhi strains were calculated previously [21]. For all other strains, the MIC was determined by the standard broth microdilution method from the Clinical and Laboratory Standards Institute to evaluate the susceptibility of each strain (Appendix A) [22]. To prepare antibiotic solutions, ciprofloxacin (17850, Honeywell Fluka, Switzerland), kanamycin (J17924-14, Fisher Scientific, Waltham, MA, USA), and ampicillin (a9518, Sigma-Aldrich, St. Louis, MO, USA) were independently dissolved in PBS at 1500× the MIC. Cefepime (J6623703, Fisher Scientific, Waltham, MA, USA) was dissolved in DMSO due to its low solubility and used at a concentration of 0.2 mg/mL.

### 2.3. Planktonic Persister Assays

Strains were inoculated in 3 mL TSB and grown on a roller drum overnight at 37 °C. The following day, the strains were removed from the incubator and normalized to an OD_600_ value of 0.4 (ND-1000, Nanodrop, Fisher Scientific, Waltham, MA, USA), representing ~2 × 10^9^ cells/mL. Subsequently, 100 μL of normalized solution was added to 2900 μL of LB +/− 1% bile and grown overnight, again, at 37 °C. Following overnight growth, 1 mL of each sample was moved to a new 1.75 mL centrifuge tube. Samples were spun at 5000 rpm for 5 min at room temperature. Media were removed and the pellet was resuspended in 1 mL of PBS with or without (control) antibiotic. Samples were incubated at 37 °C on a nutator for 24 h. Following this incubation with antibiotics, tubes were spun at 5000 rpm for 5 min at room temperature. Media were removed and the pellet was resuspended in 1 mL of sterile PBS. Samples were then serial diluted and drip plated on LB plates for CFU enumeration.

### 2.4. Biofilm Persister Assays

To prepare for biofilm assays, 96-well polystyrene plates were coated with a cholesterol (C8667-256, Sigma-Aldrich, St. Louis, MO, USA) solution consisting of 5 mg/mL cholesterol dissolved in a 1:1 solution of ethanol:isopropanol (459828-1L, 34863-L, Sigma-Aldrich, St. Louis, MO, USA). Subsequently, 100 uL of the cholesterol solution was added to each well and allowed to evaporate overnight while nutating (88-861-041, Fisher Scientific, Waltham, MA, USA) in a sterile biosafety cabinet. Strains were inoculated in 3 mL TSB and grown on a roller drum overnight at 37 °C. The following day, the strains were removed from the incubator and normalized to an OD_600_ value of 0.4 (~2 × 10^9^ cells/mL). Following this, 198 μL of fresh TSB +/− 1% (human/ox) bile was added to the cholesterol-coated 96-well plates and 2 μL of the normalized culture was added to each well. Inoculated plates were incubated at 30 °C while nutating for 5 days. Media were carefully removed every day and 200 μL of fresh TSB +/− 1% (human/ox) bile was added. Following 5 days of growth, media were removed and wells were rinsed with 200 μL of sterile PBS. The PBS was removed and 200 μL of antibiotic solutions was added to each well, as well as PBS for the controls. The plate was incubated at 30 °C while nutating for 24 h. Following incubation with antibiotics, wells were scraped and moved to micro centrifuge tubes. Tubes were spun at 5000 rpm for 5 min at room temperature. Media were removed and the pellet was washed in 1 mL of sterile PBS. Samples were then resuspended, serial diluted, and drip plated on LB plates for enumeration.

For the multiple antibiotic exposure experiment, biofilms were prepared and treated with antibiotic (or PBS control) as described above but incubated (at 30 °C, nutating) for 3 days. Six sets of replicates for each condition were established in 96-well plates. After each round of antibiotic (or PBS control), one set of replicates was plated for CFU enumeration (as describe above). The remaining sets were incubated again in media (TSB or TSB + HB) for 3 days, and the cycle was repeated until the final set of replicates was treated with antibiotics for a total of 5 rounds (Figure 5A).

### 2.5. Statistical Analysis

All experiments were performed with three or more biological replicates and repeated at least three times. Only groups with comparable treatments were statistically analyzed. Raw CFU numbers are provides in Appendix A. Total cell counts were log transformed and compared using a 2-way ANOVA with Tukey post-hoc testing with the statistical software GraphPad Prism version 7.4.0. Statistical significance was represented as follows: ns, not significant; *, *p* ≤ 0.05; **, *p* ≤ 0.01; ***, *p* ≤ 0.001; and ****, *p* ≤ 0.0001.

## 3. Results

### 3.1. Effects of Bile in Persister Formation of Various Salmonella Populations

*Salmonella* can be found in the GB lumen, epithelia, and on the surface of GS [3,23]. To model planktonic cells in the lumen, we grew *Salmonella* strains on liquid TSB media with and without bile (control). *S*. Typhi and *S*. Typhimurium are used in all assays because both can establish chronic GB infections involving biofilms [3]. These strains were treated with antibiotics at 1500 times their MIC (Appendix A) for 24 h. For persister formation assays, we chose the antimicrobials ciprofloxacin and kanamycin. The first was selected because it is the treatment of choice as a first-line therapy for children and adults in the United States [24], the latter was chosen because it has a different cellular target than ciprofloxacin. For *S*. Typhimurium, there was a stark contrast between the two antibiotics used: for ciprofloxacin, the number of persisters was below 0.01% in all conditions (Table 1), while the percentage of persisters recovered after kanamycin treatment was much higher (4.61% in TSB, 1.18% in HB, 25.8% in OB). Interestingly, the number of persisters after ciprofloxacin treatment in human bile increased slightly with respect to rich media alone, but the difference was not significant (Table 1, Figure 1). *S*. Typhi displayed a considerably higher percentage of persisters in ciprofloxacin than *S*. Typhimurium in TSB (0.12%), TSB + HB (0.07%), and TSB + OB (0.12%). Kanamycin had a comparatively negative effect on the number of *S*. Typhi persisters recovered, as the percentage was much lower than *S*. Typhimurium in all conditions (TSB, 0.69; TSB + HB, 0.02%; TSB + OB, 0.06%). Notably, both types of bile were significantly detrimental to persister formation after kanamycin treatment in *S*. Typhi (HB, *p* < 0.0001; OB, *p* = 0.0011).

To model biofilms on GS, we grew *Salmonella* on cholesterol-coated 96-well plates. Biofilms were grown for five days and treated with an antibiotic for 24 hrs. Overall, the proportion of persisters was remarkably higher when compared to planktonic growth (Table 1). For *S*. Typhimurium, no significant differences were found in the number of persister cells formed in either ciprofloxacin or kanamycin in ox bile (Figure 2A). Remarkably, the type of bile influences persister formation, as there were significantly fewer persisters after kanamycin treatment in human bile but not ox bile (Figure 2A,B; *p* < 0.0001,). *S.* Typhi behaved similarly to *S*. Typhimurium, with no differences in the proportion of persisters after ciprofloxacin treatment in either type of bile, but exhibited a difference in the number of persisters versus *S*. Typhimurium after kanamycin treatment (Figure 2; ox bile *p* = 0.0127, human bile *p* = 0.0042). Analogously to the planktonic results, for biofilms on cholesterol-coated surfaces, the combination of human bile and kanamycin is detrimental to persister formation in *S*. Typhi, with persister cells below detection limits in certain experimental sets. In summary, as expected, biofilms are a reservoir for persister formation but, contrary to our hypothesis, GB/GS conditions had little or no effect on the number of persisters recovered after ciprofloxacin treatment in either planktonic or biofilm populations. GB/GS conditions also had an overall negative effect on persister formation after kanamycin treatment, especially in *S*. Typhi.

### 3.2. Human Bile Has a Varied Effect on the Amount of Persister Cells Found after Ciprofloxacin or Kanamycin Treatment in Biofilms of S. Typhi Clinical Isolates

*S*. Typhi clinical isolates from our collection were also tested for their ability to form persister cells in human bile. In general, these isolates showed a great variability in persister formation compared to our laboratory strain Ty2, and also had lower cell numbers in the control conditions (no antibiotics, Figure 3). Two isolates showed no difference in any conditions (JSG3074 and JSG3400). Unlike the lab strains that grew equally well in the TSB and TSB + human bile controls, three clinical isolates showed a significant decrease in cell numbers when grown in TSB + human bile (JSG3441, JSG3979, and JSG 3981). For this reason, even though isolate JSG3979 showed a decrease in persister numbers after ciprofloxacin treatment, it probably reflects the lower number of bacterial cells in general found in human bile vs. the control, rather than an actual decrease in persister formation. Similar to the lab strain when grown in human bile, four of the six clinical isolates showed a greatly decreased number of persister cells after kanamycin treatment (JSG3076, JSG3441, JSG3979, and JSG3981). Very few persister cells were recovered after kanamycin treatment in two strains (JSG3074 and JSG3400) both with or without human bile. Overall, the clinical isolates behaved similarly to the laboratory strains in GS conditions. Ciprofloxacin had no effect and kanamycin a mostly negative one, although there was considerable variability between isolates.

### 3.3. The Cellular Target of an Antibiotic Influences the Amount of Persisters

Since we observed significantly different results between ciprofloxacin and kanamycin, we speculated that the antibiotic target could influence the amount of persisters recovered. Kanamycin inhibits protein synthesis and ciprofloxacin targets DNA replication. We decided to test ampicillin and cefepime, two antibiotics that target the cell wall. As can be observed in Figure 4, neither of these antibiotics affected persister formation in *S*. Typhimurium or *S*. Typhi when compared to the untreated control, except for a slight increase in the number of persisters recovered after ampicillin treatment in *S*. Typhimurium. These results confirm that the antibiotic target has a strong effect on the levels of persister formation.

### 3.4. Multiple Rounds of Ciprofloxacin Treatment in GS Conditions affects S. Typhimurium and S. Typhi Differently

To model the conditions that a *Salmonella* biofilm in the GB of a hypothetical typhoid carrier might experience with multiple antibiotic treatments over time, we set up a simple in vitro system using ciprofloxacin, the most widely used antibiotic for typhoid infections [25,26,27]. *Salmonella* biofilms grown in rich media or media plus human bile were exposed to multiple rounds of antibiotic, with a recovery time in the original growth conditions in between treatments (Figure 5A). The results are summarized in Figure 5B. *S.* Typhimurium biofilms reached peak cell numbers at round 3, which was followed by a slow decrease. Interestingly, while the total number of cells decreases over the rounds of exposure, the fraction of persisters recovered in the control (TSB) after ciprofloxacin treatment increased from 6.47% at round 1 to 57.74% at the last round. The presence of human bile had the opposite effect, as persister numbers went from 8.61 to 3.82%. For *S*. Typhi, the number of cells were steady for the first three rounds then decreased in rounds 4 and 5. Round 2 was the only time point where there was a significant difference between the number of persister cells in the different growth conditions (*p* = 0.04). The percentage of persisters in the control went down from 1.36% in the first round to 0.10% in the last, while human bile increased the proportion of persisters slightly from 2.34% to 3.13%. In summary, the above experiments in *S*. Typhimurium confirm previous findings that show repeated antibiotic treatment increases the proportions of persisters in a population, but GS conditions abolished this trend. For *S*. Typhi, the opposite was true, as GS conditions modestly increased the fraction of persisters.

## 4. Discussion

The number of persisters recovered after ciprofloxacin treatment in rich media in this study (Figure 1) was in accordance with the majority of published planktonic *Salmonella* studies [17] but lower than others [28,29], possibly due to the high concentration of anti-microbial used here (1500× MIC). As stated above, it is believed that one of the main reasons chronic biofilm infections are so recalcitrant is because biofilms can provide protection to persister cells [12]. In general, our results indicate that this is true for both *S*. Typhimurium and *S*. Typhi (Figure 1 and Figure 2, Table 1). This was especially marked for *S*. Typhimurium after ciprofloxacin treatment, where the difference in the proportion of persisters was several logs (Table 1). Remarkably, this was not true in human bile, where the proportion of persisters in planktonic and biofilm populations were almost identical (0.01 vs. 0.04%, Table 1). This is in agreement with previous studies where *Salmonella* biofilms showed a higher level of persistence than planktonic cultures [28]. In this respect, *Salmonella* seems to behave differently to other important pathogens, such as *P. aeruginosa* and *S. aureus* that show similar levels of persister cells in biofilm and planktonic cultures in the stationary phase [30,31].

A high variability in the levels of persistence has been widely documented [32,33]. A recent review article combined persister data from different studies on natural isolates of *E. coli* and found that persister levels in different antibiotic classes could vary from 0.00001% to 98% [33]. This variability was observed in our experiments using *S*. Typhi clinical isolates (Figure 3). The clinical isolates exhibited different responses to growth conditions and antibiotic treatments, with the fraction of persisters varying from strain to strain. The clinical isolates, similar to the lab strain, show high levels of persisters after ciprofloxacin treatment when grown as a biofilm. Additionally, all clinical strains show a high sensitivity to kanamycin. In concordance with our work, a recent study also found high heterogeneity in the level of persisters found after ceftazidime and ciprofloxacin treatment in planktonic and biofilm cultures of different *S. enterica* serovars [28]. Further, treatment with antibiotics such as cefotaxime, azithromycin, and ciprofloxacin has been shown to select for mutations detrimental to biofilm formation [34], which could partly explain the high variability in biofilm formation and persistence seen in our clinical isolates. It is important to note that given our limited number of clinical isolates it is not possible to draw broad conclusions.

One of the most interesting findings in our study was the decrease in persister cells in *S*. Typhi when treated with kanamycin and grown in bile. This phenomenon was also observed in *S*. Typhimurium, but only in biofilms grown in the presence of human bile (Figure 1, Figure 2 and Figure 3). Since this behavior was observed with kanamycin, which targets protein synthesis, but not with ciprofloxacin, which targets DNA replication, we hypothesized that the target of the antibiotic plays an important role in this reduction in persister cells. To further investigate, we decided to test ampicillin and cefepime, antibiotics that target the cell wall. As can be observed in Figure 4, these antibiotics do not induce a decrease in persister cells formation, as both *Salmonella* species are highly tolerant to these antibiotics, with only *S*. Typhimurium showing a slight increase in persister formation after ampicillin treatment in human bile. Antibiotics that target cell-wall biosynthetic enzymes are not very effective against slow or non-growing cells [35] such as those found in biofilms. Additionally, these antibiotics can enrich these slow or non-growing cells [36]. The reduction in persister cells in kanamycin could be a result of the arrest of protein synthesis involving protein aggregosomes, which has been reported to facilitate bacterial cell dormancy in *E. coli*. These aggregates were significantly diminished after treatment with the protein synthesis-blocking antibiotic chloramphenicol [37]. Alternatively, *Salmonella* persister cells could require a protein to survive in bile, and treatment with kanamycin could block the synthesis of this protein, making dormant cells vulnerable to bile. For example, *S*. Typhimurium persisters are known to be capable of remaining metabolically active, even in their dormant state. In macrophages, persisters are induced under the acidic conditions of the intracellular vacuole [38] and can reprogram host macrophages in vivo using their partially active effector delivery system [39]. This delivery system would be inactive if protein synthesis was blocked.

It has been reported that periodic and repetitive treatment of patients with antimicrobials commonly leads to hyperpersistance, where mutations that favor persistence are accumulated [40,41,42,43]. Furthermore, bacterial persistence has been shown to adjust rapidly to antimicrobial treatment frequencies, driving swift evolutionary adaptation [44]. In order to simulate a hypothetical condition where a chronic typhoid patient would receive several rounds of ciprofloxacin treatment, we set up an in vitro experiment where *Salmonella* biofilms were grown in conditions where bile was replenished, as it is in the GB [45], and then allowed to grow for a period of time before treatment with ciprofloxacin, with this cycle repeating five times. The results, summarized in Figure 5, indicate that there is a steady decline in total cell numbers after the continued rounds of antibiotic treatment in GB conditions. This was not expected but perhaps not surprising, as the continued onslaught of fresh bile plus antibiotic is a very challenging treatment. As the literature reports [40,41,42,43], the percentage of persisters in rich media (control) increased dramatically for *S*. Typhimurium, from 6.47 to 57.74%. Unexpectedly, bile had the opposite effect, reducing the percentage of persisters from 8.61 to 3.82%. *S*. Typhi displayed a more modest but opposite response; in the control conditions, persisters decreased from 1.36 to 0.1% but presented a modest increase from 2.34 to 3.13% in GS conditions. This could possibly reflect the evolutionary process that *S*. Typhi has undergone as a human pathogen, using the GB and GS as a niche for persistence.

## 5. Conclusions

As expected, biofilms provided a niche for persister formation; however, contrary to our hypothesis, the adverse conditions of the GB did not “prime” *Salmonella* populations for increased persister cell formation, but rather were detrimental to persisters. However, this phenomenon does not seem to be universal and can vary between serovars, strains, and antibiotic families. More work will be required to understand the complex dynamics between planktonic and biofilm communities in the GB and their potential for persister cell development. We cannot rule out that the activity of a yet unidentified factor that contributes to persister cell survival in bile is rendered ineffective by kanamycin treatment. Additionally, it is possible that the slight but detectable increase in persister cell percentages under repeated ciprofloxacin treatments in GS conditions for *S*. Typhi could become significantly larger with additional rounds of antibiotic treatment. Understanding these complex mechanisms will bring us closer to the goal of eradicating typhoid fever.

## Figures and Tables

**Figure 1 microorganisms-10-02276-f001:**
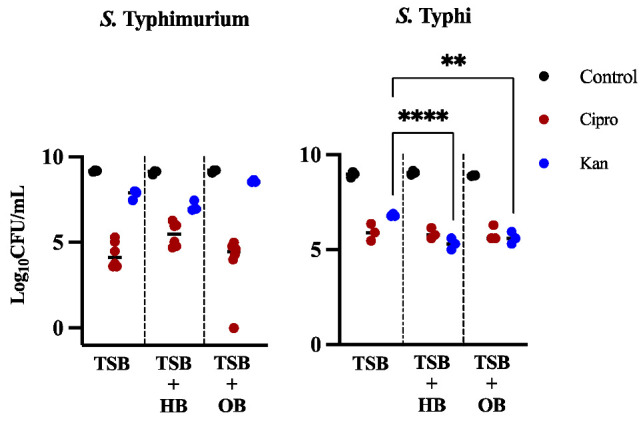
Persister cells recovered in *S*. Typhimurium 14028 and *S*. Typhi Ty2 laboratory strains after treatment with ciprofloxacin (red circles) or kanamycin (blue circles) in planktonic cells growing in ox bile (OB) or human bile (HB) versus total cells (strains not treated with antibiotics are represented by black circles). Representative charts are shown from one experiment performed with four or more biological replicates and repeated at least three times with similar results. A one-way ANOVA followed by Tukey’s multiple-comparison test were used to compare the different conditions (**, *p* ≤ 0.01; ****, *p* ≤ 0.0001).

**Figure 2 microorganisms-10-02276-f002:**
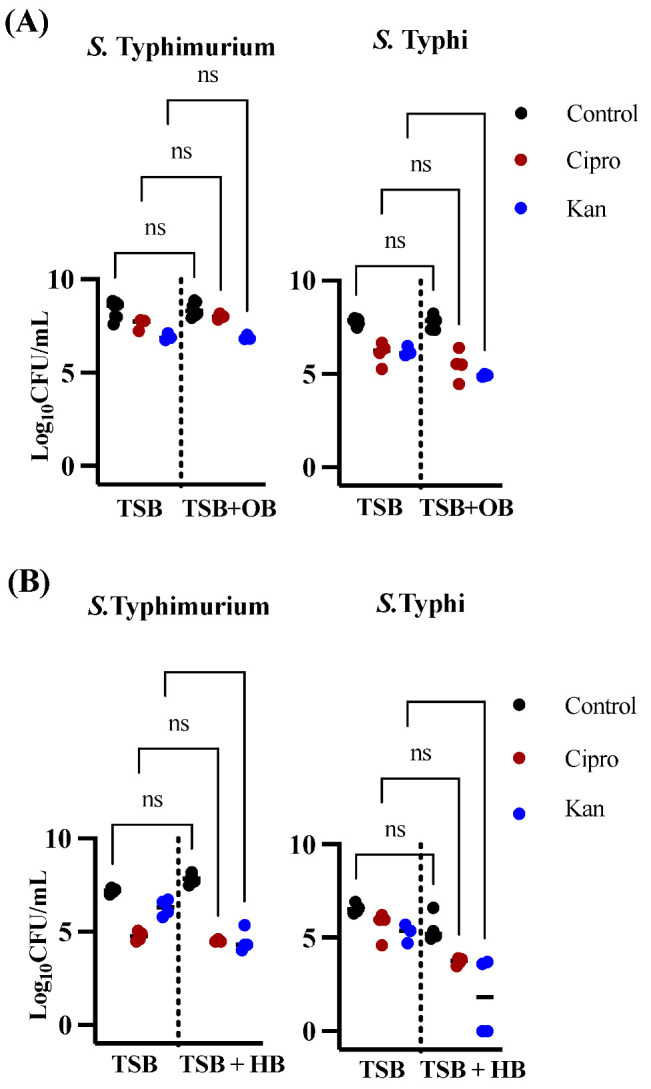
Persister cells recovered in *S*. Typhimurium 14028 and *S*. Typhi Ty2 laboratory strains after treatment with ciprofloxacin (red circles) or kanamycin (blue circles) in biofilms growing in ox bile (**A**) or human bile (**B**) versus total cells (strains not treated with antibiotics are represented by black circles). Representative charts are shown from one experiment performed with four or more biological replicates and repeated at least three times with similar results. A one-way ANOVA followed by Tukey’s multiple-comparison test were used to compare the different conditions (ns, not significant).

**Figure 3 microorganisms-10-02276-f003:**
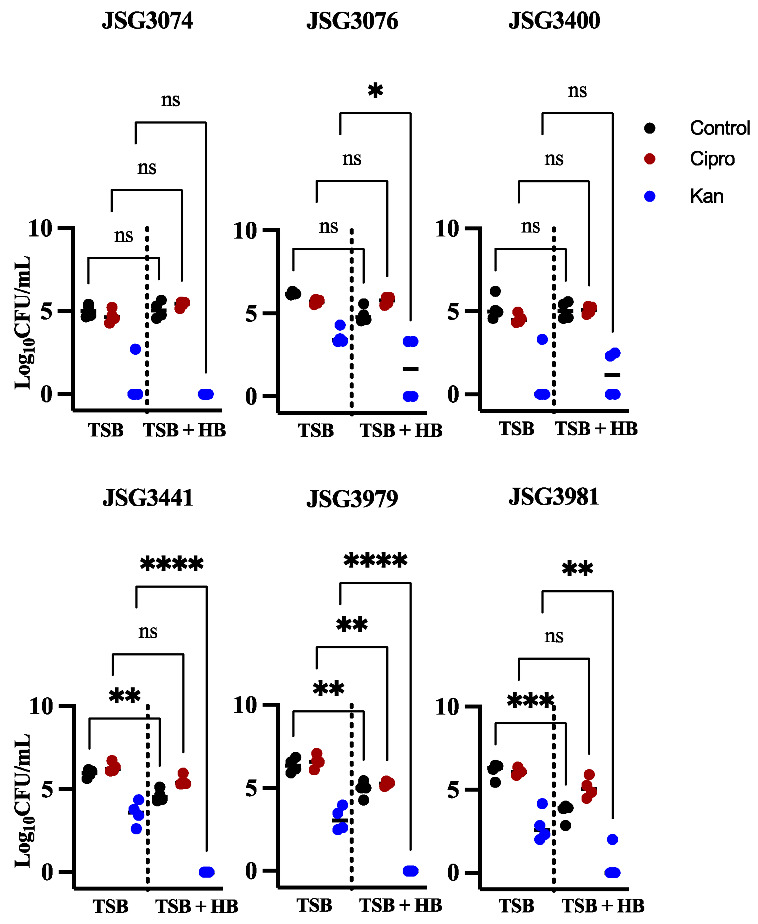
Persister cells recovered in *S*. Typhi clinical strains after treatment with ciprofloxacin (red circles) or kanamycin (blue circles) in biofilms on human bile versus total cells (strains not treated with antibiotics are represented by black circles). Representative charts are shown from one experiment performed with four or more biological replicates and repeated at least three times with similar results. A one-way ANOVA followed by Tukey’s multiple-comparison test were used to compare the different conditions (ns, not significant; *, *p* ≤ 0.05; **, *p* ≤ 0.01; ***, *p* ≤ 0.001; ****, *p* ≤ 0.0001).

**Figure 4 microorganisms-10-02276-f004:**
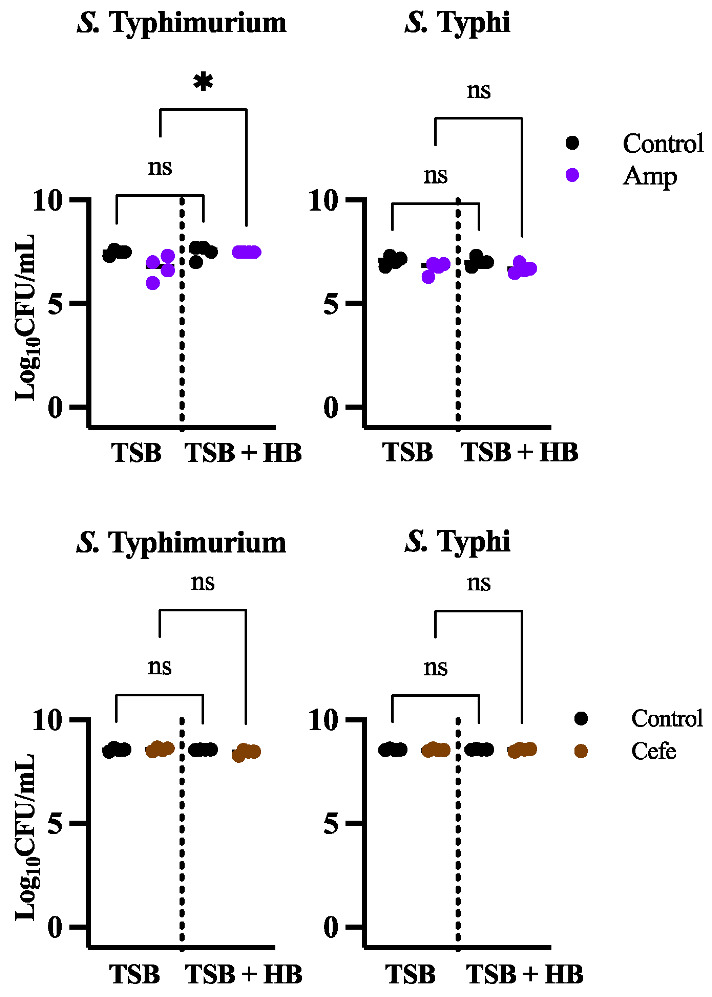
Persister cells recovered in *S*. Typhimurium 14028 and *S*. Typhi Ty2 laboratory strains after treatment with ampicillin (purple circles) or cefepime (orange circles) in biofilms growing in human bile (HB) versus total cells (strains not treated with antibiotics are represented by black circles). Representative charts are shown from one experiment performed with four or more biological replicates and repeated at least three times with similar results. A one-way ANOVA followed by Tukey’s multiple-comparison test were used to compare the different conditions (ns, not significant; *, *p* ≤ 0.05).

**Figure 5 microorganisms-10-02276-f005:**
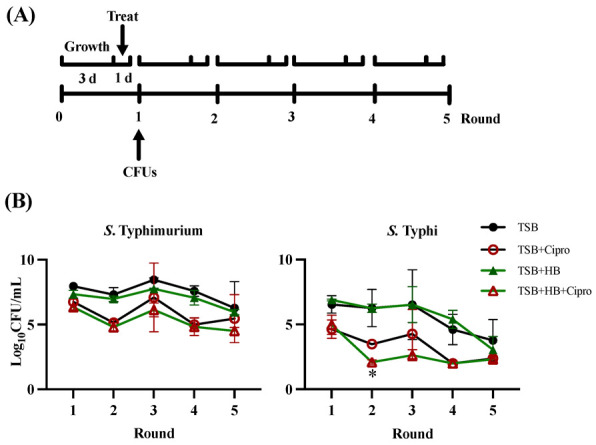
Multiple rounds of ciprofloxacin treatment on biofilms. (**A**) Experimental set-up; growth, treatment, and CFU counts for rounds 2–5 were performed the same as for round 1. (**B**) Colony forming units recovered for *S*. Typhimurium 14028 and *S*. Typhi Ty2 laboratory strains in different growth conditions after every round of ciprofloxacin treatment (red symbols) vs. control total cell numbers (black and green symbols). Representative charts are shown from one experiment performed with four or more biological replicates and repeated at least three times with similar results.

**Table 1 microorganisms-10-02276-t001:** Percentage of persister cells recovered after anti-microbial treatment in different growth conditions.

		% In Planktonic		% In Biofilm	
Organism	Media	Ciprofloxacin	Kanamycin	Ciprofloxacin	Kanamycin
*S*. Typhimurium	Control	3.07 × 10^−4^	4.61	4.89	13.73
	HB	0.01	1.18	0.04	0.08
	OB	6.67 × 10^−4^	25.78	21.55	6.70
*S*. Typhi	Control	0.12	0.69	12.16	4.12
	HB	0.07	0.02	0.52	0.20
	OB	0.12	0.06	1.67	0.08

## Data Availability

Data can be found at: https://www.dropbox.com/sh/kbtcdlm7x3sgk44/AADeI1biI20ROtPbSr05dLg0a?dl=0.

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
