# Peer review of "The Effect of the Gallbladder Environment during Chronic Infection on Salmonella Persister Cell Formation"

_microorganisms, 2022, doi:10.3390/microorganisms10112276_

Round 1

Reviewer 1 Report

This manuscript establishes phenomenon of Salmonella persister under antibiotics that leads to chronic infection in gallbladder. However, the study is required to improve many parts of manuscript such as experiment design, hypothesis, introduction and presentation of data. I would like to summarize comments below.

1. Authors need to clarify why they use ciprofloxacin and kanamycin for the study in the introduction or the first part of results.   

2. To test persistence, authors use 1500XMIC. However, this concentration is too high, because we cannot use 1500XMIC in the clinic to control bacteria infection. Thus, authors need to use reduced concentrations of antibiotic, which is related to clinical treatment such as 3X or 5X MIC. 

3. I could not find asterisk or NS marks according to statistic analysis in Figures. Authors should clarify and check that they present statistic analysis in the Figures. 

4. In the results section, authors did not mention or summarize about meaning of their data. This is important to follow the manuscript and logical presentation of data. I would like to suggest to add summary at the end of paragraph in the result section. 

5. Authors mentions high heterogeneity in biofilm and persistence of clinical isolates. However, they did not measure heterogeneity of persistence. Authors determined persister numbers of bacteria after antibiotics treatment. This did not indicate heterogeneity of persistence. 

Author Response

  1. Authors need to clarify why they use ciprofloxacin and kanamycin for the study in the introduction or the first part of results.   

This has been added in the results section.  See lines 152-155 which state, “For persister formation assays we chose the antimicrobials ciprofloxacin and kanamycin, the first was selected because it is the treatment of choice as a first line therapy for children and adults in the United States  (24), the latter was chosen because it has a different cellular target than ciprofloxacin.”

  1. To test persistence, authors use 1500XMIC. However, this concentration is too high, because we cannot use 1500XMIC in the clinic to control bacteria infection. Thus, authors need to use reduced concentrations of antibiotic, which is related to clinical treatment such as 3X or 5X MIC. 

The recently published Definitions and Guidelines for Research on Antibiotic Persistence (PMCID: PMC7136161) recommends testing persistence at concentrations considerably higher than the microbe’s MIC, and a 1500X concentration has been used by others in the field studying persistence. Additionally, biofilms are innately tolerant to antibiotics and have been reported to be 100 to 1000-fold more resistant than planktonic cells. Given these criteria we felt that, since our study was meant to be an in vitro analysis of persister cells, choosing a very high concentration would dissipate any confusion between tolerance and persistence in our biofilm assays. Finally, to compare biofilms to planktonic cells, we used the same concentration on both populations.

  1. I could not find asterisk or NS marks according to statistic analysis in Figures. Authors should clarify and check that they present statistic analysis in the Figures. 

We are a bit surprised by this comment as the figures did possess the applicable statistics, asterisks, ns, etc. We only calculated statistical significance between comparable groups (for example TSB vs TSB+Bile, TSB+Kan vs TSB+bile+Kan, and TSB+cipro vs TSB+bile+cipro) so only these comparisons have the ns or asterisk markings in the figures. This has been specified in the Statistical analysis section of Materials and Methods, and the P values are described in the figure legends. See lines 140-1 and 143-4.

  1. In the results section, authors did not mention or summarize about meaning of their data. This is important to follow the manuscript and logical presentation of data. I would like to suggest to add summary at the end of paragraph in the result section. 

 Summary sentences have been added at the end of each section in the Results sections.

  1. Authors mentions high heterogeneity in biofilm and persistence of clinical isolates. However, they did not measure heterogeneity of persistence. Authors determined persister numbers of bacteria after antibiotics treatment. This did not indicate heterogeneity of persistence. 

           The reviewer is correct, we meant that the effects we found varied significantly between serovars and strains. The word “heterogenicity” has been replaced with “variability”.

Reviewer 2 Report

Dear editor and authors,

Thanks for the opportunity to review this work. The paper is about a relevant topic. It has scientific relevance, it was well conducted, using strains of isolates from different places with antibiotics using for possible treatments. The number of isolates is not large, which is a limitation of the study, which should be included in the discussion/conclusion of this paper.

About the title, if the authors consider it valid, the antibiotics used in part of the study could be inserted, but this is just a suggestion.

Introduction

Introduction is appropriate written, with good references that validate the study. The references are not very up-to-date, but they are sufficient to support the purpose of the study.

Materials and methods

It is well described.

Questions:

Were these analyzes performed in replicates? If yes, how many? If they were not done, it could be one of limitations of this study, and it should be clear in the text.

In addition to replicates, were the same assays performed on different days? This would make the results more robust.

Why were these antibiotic concentrations chosen for the field strains? Make it clearer, because in the discussion the amount of ciprofloxacin may have been an important factor for the result. This can be done in the discussion.

Results

Be careful not to put discussion at this step  (i.e. first sentence of results).

I found the results well described, with necessary information in the graph. I suggest putting the raw data as supplementary material, along with replicates.

Discussion/Conclusion

The discussion was based on current articles, and limitations that the study presents were placed. It would be interesting to make it clear that robust conclusions cannot be drawn due to number of isolates used. If the tests have not been done in triplicate, nor done on different days, this must also be taken into account as a limitation and made clear in the text.

Author Response

About the title, if the authors consider it valid, the antibiotics used in part of the study could be inserted, but this is just a suggestion.

Thank you for the suggestion but since we used four antibiotics, we think it is best to leave the title as written.

Introduction

Introduction is appropriate written, with good references that validate the study. The references are not very up-to-date, but they are sufficient to support the purpose of the study.

Materials and methods

It is well described.

Questions:

Were these analyzes performed in replicates? If yes, how many? If they were not done, it could be one of limitations of this study, and it should be clear in the text.

Every experiment was done with at least 3 replicates and each experiment was performed at least 3 times on different days. We have added a more detailed description of replicates and number of experiments in the Statistical Analysis section of the Materials and Methods. See line 140.

In addition to replicates, were the same assays performed on different days? This would make the results more robust.

            All experiments have been done at least three times on different days.

Why were these antibiotic concentrations chosen for the field strains? Make it clearer, because in the discussion the amount of ciprofloxacin may have been an important factor for the result. This can be done in the discussion.

We calculated the MIC for every strain then multiplied this value by 1500 and used this as our working concentration for all assays. The concentration is high because we wanted to make a clear distinction between persistence and other antimicrobial survival strategies (like tolerance or genetic resistance). We have included an explanation in the results section. See lines 152-155.

Results

Be careful not to put discussion at this step  (i.e. first sentence of results).

We think this sentence should remain because it explains why we tested both planktonic and biofilm populations in our assays, since both can be found in the gallbladder environment.

I found the results well described, with necessary information in the graph. I suggest putting the raw data as supplementary material, along with replicates.

The raw data is now included in a spread sheet in the supplementary materials

Discussion/Conclusion

The discussion was based on current articles, and limitations that the study presents were placed. It would be interesting to make it clear that robust conclusions cannot be drawn due to number of isolates used. If the tests have not been done in triplicate, nor done on different days, this must also be taken into account as a limitation and made clear in the text.

We have added this caveat in the Discussion section. See lines 312-3.

Round 2

Reviewer 1 Report

I agree to accept current version of manuscript.